# Socio-Problematization of Green Chemistry: Enriching Systems Thinking and Social Sustainability by Education

**Leonardo Marcelino** [1,2,*] [ID]**, Jesper Sjöström** [1] [ID] **and Carlos Alberto Marques** [2]

1   Department of Natural Science-Mathematics-Society (NMS), Malmö University, SE-205 06 Malmö, Sweden; jesper.sjostrom@mau.se
2   Departamento de Metodologia de Ensino, Universidade Federal de Santa Catarina, Florianópolis-SC 88040-900, Brasil; carlos.marques@ufsc.br
*   Correspondence: leonardo-victor.marcelino@mau.se

**Abstract:** The current research on systems thinking criticizes the additive nature of green chemistry (GC) not being supportive of systems thinking to achieve holism in its practices. This paper argues that systems thinking should comprise of the social issues, and, therefore, it studies renowned papers by GC pioneers and reviews on the field regarding how they address the social dimension of sustainability. It points out how GC has ignored social sustainability in its discourses, practices, and evaluations, leading to a reductionist interpretation of sustainability. Then, this paper presents some challenges to be overcome in order to achieve balanced sustainability. A systemic chemical thinking is advocated, considering chemistry in culture and chemistry as culture, expanding the chemistry rationality from ontological and technological dimensions into the epistemological and ethical ones. It is then discussed how chemistry education can help to promote sustainability in a broad and systemic way.

**Keywords:** green chemistry; bildung education; social participation in science; democracy in science; social sustainability; chemistry education; chemical thinking

## 1. Introduction

A quarter of a century after the emergence of green chemistry (GC), it has been verified that its practices towards sustainable chemistry may be more incremental than transformative if the twelve principles are not considered as a cohesive system setting the hows and whys of chemistry practice [1–3]. One of the causes of this problem is the lack of systems thinking in GC [2], which leads to chemical practices that focus on specific aspects and do not address the complexity of the unsustainability issue [3,4].

Following the Responsible Care Program, an autonomous initiative of large chemical corporations initiated in 1989, GC rose as an academic and industrial pursuit of economically viable technological innovation to reduce waste emission, prevent pollution and minimize health hazard in chemical activities. Some positive correlation between GC and sustainability has been verified, but this relationship has not always been clear, and the role of GC is not explained [5]. Concerns have arisen from the genesis of GC amidst the industrial system, and from its definition, which emphasizes the compatibilization of economy and environment [6], as stated by Anastas and Beach, "It has been said that the term green chemistry was derived from the dual connotations of the word "green" concerned with the environment and the color of the US dollar. There is an additional connotation of the word "green," which is young, fresh, and new" ([7], p. 20).

Chertow [8] and Commoner [9] proposed that environmental impact is a function of three factors—population size, consumption (or affluence), and technological level—and, therefore, sustainability is based on three pillars (triple bottom line model): society, environment, and economy [10]. At the Stockholm Conference 1972 [11] and in its The Limits to Growth report [12], economic growth and environmental preservation are said to be incompatible, although the concept of sustainable development presented in the Brundtland Report, fifteen years later, denies this incompatibility, promoting technological development as a means to overcome it [13].

When action is carried out, society may be invited to monitor and evaluate its performance and impacts. Because the action always has a degree of unpredictability, it is necessary to monitor its results. On the basis of these considerations (the need to balance the economy, society and the environment, and the tendency of the concept of sustainable development in the Brundtland Report to focus on the economy at the expense of the environment and society), the relationship between GC and sustainability is put to question. The main interest of this paper is to discuss how the social dimension appears in GC research, and reflect on how education can contribute to a more complex and systemic vision of what sustainability is, and the role of chemistry in achieving it.

Quality education is defended as the fourth Sustainable Development Goal proposed by UNESCO as a means to achieve social sustainability (ending inequality in access to education and promoting training to better jobs) and as a means to develop a mentality towards sustainability [14,15]. Chemistry deals with the manipulation of matter and energy to create useful products for society, and therefore it has a major role in promoting sustainability [16,17], and chemistry education should focus on this aim. Anastas and Zimmerman argue that GC is the chemistry of sustainability and that enabling system conditions is a conceptual framework of GC. Nonetheless, several authors propose that systems thinking should be incorporated in chemistry education [17–19], improving the quality of education and promoting sustainability.

The next section discusses social sustainability, its definition, limits, and assessment. The Section 3 analyzes how social participation in scientific endeavors is possible and why it should be done. Sections 4 and 5 present methods and results, respectively. Finally, Section 6 advocates for a chemistry education that allows a complex understanding of sustainability in its multiple dimensions, which provides for an understanding of (green) chemistry as culture and within culture, and that aims at full human development.

## 2. Social Sustainability: Definition and Assessment

Social sustainability is a complex concept where the basic idea is that there is something in society that must be maintained and sustained. However, defining what this ideal state of society is and how to assess it is a complex task. In a compilation on the understandings of the social dimension of sustainability, Dillard, Dujon, and King [20] use a work definition of social sustainability that considers it in two aspects: as a means to achieve environmental and economic sustainability, and as an end to an action, aiming at well-being and social health. Social sustainability is a necessary condition and an end for sustainability in general, which allows us to infer that, without social development, there is no way to reduce environmental impacts and promote economic development, hence its importance for GC as well.

Boström [21] analyzes the difficulties in conceptualizing social sustainability, identifying in the literature a confusion between substantive aspects (what is social sustainability) and procedural aspects (how to achieve social sustainability). The author [21] argues that the substantive aspect of the social dimension has to do with quality of life, well-being, and happiness as well as with the fulfillment of basic needs: food, home, income, fair distribution of advantages and disadvantages, rights equality, access to social and environmental infrastructure, opportunities for learning and self-development, security, health, social cohesion, cultural diversity, respect to traditions, sense of belonging and social recognition. Social welfare is achieved through the procedural aspects of social sustainability, which involves broad participation in decision-making and regulation of scientific

and technological activities, e.g., access to information about the project (transparency), participation in the decision-making throughout the project, proactive communication between the stakeholders, empowering participation (education, divulgation, and economic compensation), participation in defining the assessment criteria and indicators, and social monitoring of the working project. In other words, what must be achieved and sustained is the welfare of society, and the path for that it is promoting autonomy, competence, and relatedness [22,23].

Promoting sustainability is a complex problem, and social sustainability is perhaps its most difficult tenet to be promoted and analyzed. Sustainability assessments have disregarded the social dimension [21] because of the complexity of establishing and measuring social indicators [24–26]; the high stakes of social sustainability and the difficulty in conciliate it to the environment and the economy; the close relationship between the historical and institutional development of sustainability, on the one hand, and environmental sustainability, on the other; the inefficacy of market-based strategies to promote social justice; and lack of attention to the relationships, synergies, and trade-offs between the three tenets, usually regarded as independent dimensions.

The United Nations Environment Program (UNEP) and the Society of Environmental Toxicology and Chemistry (SETAC) propose the social life cycle analysis (SLCA) as a suitable methodology for analyzing social impacts (negative or positive consequences on the welfare of stakeholders) arising from an activity [27]. However, the indicators for SCLA are not clear and vary in the depth and breadth of its consideration of the social dimension.

From an ethical or normative point of view, SLCA is not efficient in determining whether a process should be carried out or not [28]. Its assessment may indicate whether an action may have impacts and to what degree, but it is not enough to determine whether the action is really necessary. Hence, there is a need to actively involve multiple stakeholders during the entire process of design, implementation, and running of a technological enterprise [21,27]. Broad social participation helps to curb enthusiasm (technological optimism) by redefining what is possible and what must be compromised, to explain the political nature of sustainability where the aims of society are democratically discussed, and to reinterpret the environment in light of social justice to embody the urban environment, the discussion on racism and gender equality, home access, poverty and so on [21].

It is also needed to approach social sustainability as a complex concept with a different epistemological nature than the environment or the economy, requiring research conducted jointly by social and natural scientists in order to fully develop its fundaments as well as to find ways to integrate and compare the three dimensions of sustainability regarding their intrinsic differences, and avoiding applying the same assumptions on them [28]. Democratization, in the debate about the concept and how to assess it, is a condition that determines the success of social sustainability and is also a valued attitude in any process.

## 3. Social Participation in Science

In the past, when greater social participation in science and technology was proposed, it was common to hear the following provocation: How can society decide between a continuous flow or batch process? How can the layman decide between toluene or water as a solvent? These are valid points, but not altogether unassailable, as their speakers consider chemistry only as a hypothetical technical activity detached from the real world. The following questions may be added: What problem does a given action aim to answer? What solution is being proposed? Who are the beneficiaries, who is excluded, who is harmed? Is this process necessary, good, or fair? Here, chemical activity is set in the real world; chemistry is not a mere theoretical activity (only explaining what the world is), but a practical one, that intervenes in the world and transforms it [29–31].

If a chemical activity is viewed as purely theoretical, without any relevant application to society or any environmental or social impact, that will deny the very history of chemistry and its responsibility. The fact is that chemistry produces materials, interferes with the world, engages in problem-solving, reviews its practices, as well as theoretically researches the world. The intention or desire to practice

chemistry is not exclusive to scientists; it may be a collective desire of society. The use of the verb may underscore the need not to presume the wishes of society, but it highlights the importance of investigating them systematically.

A practical activity is born by desire, but it is implemented by deliberation [32], i.e., by collecting and analyzing all the possible strategies, tools, and methods to achieve the goal. At the deliberation stage, it is important to recognize that solving a practical problem requires more than knowing how to apply universal knowledge (the laws of science) because science is built on models and idealizations of the world [33]. Practical problems are idealizable, not ideal; its numerous variables cannot be forcibly eliminated [32]. Thus, experts in only one branch of knowledge are unable to make a broad survey of possible alternatives to achieve the previously formulated desire. Also, understanding what the problems are and the best way to solve them requires acknowledging the needs, desires, and practices of lay people. Dialogue among different social groups is necessary.

The great contribution of science lies in its ability to systematically investigate the possible alternatives arising in the deliberation stage. From this space of alternatives created with broad participation of society, it is possible to evaluate which means are effective and most efficient. However, society is also required to debate their preferences and the acceptable types and patterns of risks inherent in the process. It is necessary to discuss with the population which risks they consider acceptable or intolerable in a given context if the action should prioritize speed in results or low costs (efficiency issues), especially taking into account the people who will be directly impacted.

Some cases make this point clear. According to Irwin [34], the measures to control the health crisis caused by bovine spongiform encephalopathy (mad cow disease) were imposed without any dialogue with the slaughterhouse workers who were to put them into practice, who regarded the experts' recommendations as unrealistic or without any practical sense. Mats Utas' [35] research on Ebola control failure as a result of expert determinations that ignored the context of vulnerable populations, and imposed regulations that the population did not understand the need for and which, sometimes, defied their own beliefs and culture. Michel Callon [33] reports one mode of relationship between science and society, where lay people act as co-producers of knowledge, being a concerned group—a collective of very interested or highly impacted individuals. The author mentions patients suffering from the same rare genetic disease gathering effort to produce data about the characteristics and development of their illness when the experts themselves had given up.

Society can influence science much more than just by being responsible consumers who choose the more environmentally benign product. Society can influence scientific agenda, debating the problems that should be addressed by scientists, backing up their research, and even organizing and collecting data to enrich the investigation. As Fiorino [36] argues, democratic social participation in decision-making processes about scientific-technological enterprises is justified by instrumental, normative, and substantive arguments:

- Instrumental argument (because it works): Democratic participation enables less social resistance, greater technological adoption, popular support for entrepreneurship, and greater trust in institutions;
- Normative argument (because democracy is a value in itself): Participating in the decision-making of processes that affect public life is an inalienable characteristic of being a citizen, which is ultimately the basic characteristic of the subject living in society;
- Substantive argument (because it alters the very nature of the enterprise): Lay people have different but equally useful knowledge for decision making. The general population, especially those directly involved with a technological alternative, have more knowledge of the problems and the context in which they are inserted, their goals, and their desires. Addressing this knowledge in decision-making gives greater complexity and greater adequacy of solutions to real problems.

## 4. Materials and Methods

As discussed above, social sustainability is a necessary condition and a desired end of sustainability. It is important, therefore, to analyze how the social dimension of sustainability has appeared in the works that spread and debate GC.

We investigated 37 (13 + 24) papers regarding the role of society in the relationship between GC and sustainability. Those papers were selected according to the following criteria:

- Nineteen reviews mentioning sustainability issues were analyzed because of their importance in establishing the boundaries of the GC domain, as they are authored by pioneers of the GC field and referenced in GC textbooks (as previously discussed by Marques and Machado [5]). Thirteen papers address social sustainability and therefore are reported here;
- Twenty-four papers were carefully chosen from the Green Chemistry Journal, relating GC and social sustainability (search terms: social life cycle sustainab *; social metric sustainab *). These papers were retrieved in March and November 2019, and no period of publication was delimited.

Those publications were fully read, their mentions of the social dimension of sustainability were extracted and analyzed regarding the role of society in GC practices, either in the definition of the research agenda, the development of the research and its regulation, or in the impacts imposed on society. It was considered how societal questions influenced the motivations and grounded the problems addressed in the research, as well as how the papers approached social participation in GC, either by political means or participatory decision-making. Finally, careful attention was paid to the indicators and tools to assess the social dimension of sustainability.

The excerpts representing the relationships between GC and society were extracted and grouped into the following analytical themes:

- Society implied in social responsibility
- Compatibility between the environment and the economy, and indirect benefits to society
- Green chemistry and regulation through policies
- Lack of public participation and GC as an elite social movement
- Lack of social life cycle indicators

Below we present the results. The analytical themes were not discussed separately as they emerged interconnected.

## 5. Results

### 5.1. Green Chemistry: Environment, Economy, and Society?

Some papers acknowledge the triple nature of sustainability [3,16,37,38], but the relationship between the three dimensions (society, the economy, and the environment) is not fully established or developed, although it is said that environmental issues are the most important, and that social and economic dimensions are of secondary importance ([37], p. G72).

When papers cite the social dimension (which they seldom do), it appears in a triad: environmentally benign, economically viable, and socially responsible [39–42]. The notion of social responsibility is strong in GC [43] and has its origins in the Responsible Care Program, aimed at better environmental and safety performance, and improving the public perception of chemistry [44]. The program emerged as a response to chemical disasters in the late 20th century [40] with the initial intention, as King and Lennox [44] point out, to avoid costs for companies with lawsuits and rigid regulatory policies, which corroborates GC's intention to be a non-regulatory movement focused on the self-adherence of the industrial sector [43,45]. The program reveals that chemists acknowledge their responsibility for some major environmental impacts and will engage in actions to avert it. However, its contradiction, according to Givel [46], is trying to reconcile the interests of corporations with those

of the community while precisely intending to change the public's perception of chemistry, and also opposing to restrictive regulations of processes and products so as not to increase costs for corporations.

Society is, therefore, considered to be just consumers dependent on chemical industries to produce their goods and being influent on the economy by their behavior [3,39,47,48]; hence the importance of constructing a good image of the corporations. This can be seen in the statement [3] (p. 1950) "Those necessary interactions [for GC development] include a supply of educated and aware chemists, collaborators in the broad range of disciplines, recognition of value of sustainable products and processes by consumers, investment by businesses and venture capital, and stable funding of research" [3]. Responsible action benefits society by minimizing impacts on health and the environment [39,41,43,49], showing a limited understanding of society as a whole: humans are more than their physical bodies, and society also deals with complex relations and wills of its subjects [40]. Thus, the idea of social responsibility becomes the strongest link of GC with social sustainability, but its lack of clarity in its definition may hide a corporate maneuver to keep economic interests above environmental and social issues.

Contrary to the idea that economic growth and environmental maintenance are contradictory [12,50], GC is based on possible compatibility between these two dimensions [39,43,51,52]. Winterton [53] argued that the demand for materials and services to maintain the lifestyle of a growing population can increase the environmental impact unless technologies are created that allow for more efficient use of resources. Manley, Anastas, and Cue [39] are very straightforward in saying that trying to "balance" economic, environmental, and social dimensions will inevitably result in trade-offs, so GC should seek a synergistic interaction through technological innovation that allows the improvement of efficiency and adds value to products, serving as a differential in the growing commodities market.

Technological innovation is the great theme of GC [1,7,38,39,48,51,53,54], their basic type of action alluding to the idea that GC has an instrumental nature [6], and that technological innovations are a categorical value [55], as seen in the following statements: "The challenge facing industry and society at large is extending technological innovation in a way that is sustainable both economically and environmentally" ([43], p. 5) and "Green Chemistry is about innovation—continuous improvement" ([39], p. 745).

Nevertheless, GC's technological optimism is an illusion that can hinder the pursuit of sustainability [6]. Technologies do not work isolated but are inserted in a technological system relying on multiple conditions to work properly [56]. That is why green technologies may not be enough to fight environmental degradation if they do not see the complex technological and social systems in which the chemical industry is positioned. That is why we need to develop systems thinking [57]. It should be remembered that the technological system encompasses all the process, inputs, and outputs from the cradle to the grave (or cradle to cradle) and the cultural domains in which it occurs, as recently pointed out by Anastas and Zimmerman [16]. Hence technological innovation is not simply additive, a mere change of apparatus, but a complete revolution on the ways people interact and produce. Think of the communication and information technologies and its relation to society; even the press was involved in a great cultural transformation around the world. Systems thinking in GC should take the social embeddedness of technologies into consideration when addressing sustainability issues [57–59].

Some GC researchers are aware of this, and consider that GC is a necessary condition for sustainability that needs political support to be effectively implemented, even if they do not agree on which type of policy should be implemented. Many of them promote GC as a normative institution, being self-organized, fighting restrictive regulations (taxes, bans, and additional costs) [1,43], and proposing positive policies [16,47,48,54] (e.g., funding for research on green technologies, tax incentives for the adoption of eco-efficient technologies). On the other hand, Thornton in 2001 (p. 1231) had already considered that GC will only contribute to sustainability if "it is conceived as part of a new policy based upon precautionary [principle]," enabling the management and even the banning of entire classes of chemicals (such as organochlorides). Later investigations consider that the

regulation on the registration, evaluation, authorization and restriction of chemicals (REACH) has some background on a precautionary and restrictive approach, although it also has a technocratic aspect, relying primely on natural science data and ignoring social issues and broad social participation.

GC, nonetheless, has kept itself away from public debate, acting as an "elite movement" [45], a social movement that relies on internal transformation via the education of its academic elite, and influencing the industry with the economic advantages of green technologies. Society is seen as the consumer, affected by the new image of the "green corporations" indirectly changing the market. This elite movement has been appointed as a slow process with no guarantee of success [45]. Green chemists themselves state that positive policies are needed to influence changes in the direction of GC. Matus [40] goes further and finds that a mix of regulations (positive and negative policies) and normative institutions (principles, self-organized commitments) is much more effective in promoting changes, proposing that GC tighten its relations to organized civil society in order to facilitate knowledge transfer, increase awareness of environmental problems, reward achievements in environmental protection, highlight the gaps of industrial activity and enrich the overall process. Hence the importance of GC in relation to environmental movements and non-governmental organizations is to promote its practices and gain both social backup and a broad set of supervisors and shareholders.

It should be emphasized that current studies show that the positive/negative correlation between technological innovation and production depends on the country's economic context and of changes in the entire technological system, not just of a single technology. In the case of GC, it must be acknowledged that technological innovation does not guarantee increased productivity (wealth) or environmental protection if there is not a political, social and technological system that supports these innovations, which means that changes in both the industrial and social structures are necessary to achieve a more environmentally benign approach.

Another point of contention in the supposed compatibility between economy and environment through innovation is the intrinsic inefficiency of technologies. As postulated by the second law of thermodynamics, any process only occurs with the degradation (anergy) of a part of useful energy (exergy) and transformation of matter from a low entropy state (high organization) to a high entropy state (low organization, high stability) [5,50]. Thus, the idea of a circular (or closed) economy is inconceivable (although it has permeated GC discussions) [3,60–67], since the total energy is maintained throughout the process, but the possibility of its use is always diminished [50,68,69]. Besides the physical limits imposed on technologies, which inexorably lead to environmental degradation, there are also problems of technological efficiency, i.e., it is not always that theoretical yields are attained in practice [53]. GC can effectively act to improve this technological efficiency, but without the pretension of a zero environmental and social impact, because that would mean contradicting the Entropy Law [16].

In a recent article, Anastas and Zimmerman [16] discuss the role of GC in promoting sustainability by discussing the field's tools, strategies, and goals. This article, an exception in the group analyzed, considers the part of GC for social welfare and presents categorical objectives that override economic factors. The authors set humanitarian objectives of GC as the proactive pursuit of well-being beyond the mere diagnosis and mitigation of impacts, a chemistry that refuses to produce substances and processes that serve war, death, and either chemical or economical addiction, respecting the free access to genetic information of nature and the individual's control of his own genetic code. Science communication in clear, accessible and true language is advocated as a means of bringing the public closer to chemistry, demystifying the public perception about the field, and attracting the lay-person's trust.

However, the strategies to achieve these goals are not so clear. The authors [16] argue that externalities are considered in cost assessment and that social benefits can be incorporated into the decision process. While denying that this is monetizing social and environmental qualities, the authors understand that this can create value that confer monetary benefits, which sounds paradoxical, especially when the authors themselves agree that "profit is the almighty motivator." The discussion continues to warn of the limitation of metrics in making a complex analysis of multiple variables,

many of them qualitative (such as social and environmental), which limits the scope and possibilities for innovation.

*5.2. Green Metrics and Social Indicators*

Assessing the sustainability of GC practices remains a challenge. The metrics proposed so far focus on the efficiency of processes with respect to mass and energy (E-factor, mass intensity, atom economy); some work with intrinsic risks of chemical products (Green Star, for instance), but the evaluation of sustainability of the processes is more comprehensive, and researchers have suggested the life cycle assessment (LCA) as a viable alternative. Although LCA allows for a more holistic assessment of industrial practices, it is often reduced to economic and environmental aspects because of the lack of clarity of social indicators and effective ways on how to assess them [24–26,37,38,70]. Even the environmental indicators are difficult to measure in the proposed LCA due to lack of available data on the impacts of the production, consumption, and disposal of chemicals [71,72].

Jiménez-González et al. [73] reviews the LCA processes in the pharmaceutical industry but does not report the evaluation of the social dimension. Tabone et al. [74] correlate Green Design Principles to sustainability indicators, ignoring explicit mentions to social sustainability. Kralisch, Ott, and Geriche [71] provide a tutorial-review on LCA in green chemical processes and report only one evaluation strategy of social, economic, and environmental dimensions together—the SocioEcoEfficiency analysis (SEEbalance) developed by BASF [75]. Van Schoubroeck et al. [76] review sustainability indicators in the bio-based economy (an important branch of GC) [3] and found degrees of importance attributed to the dimension: half of the assessments were one-dimensional and related to environmental impacts; 34% were two-dimensional, regarding the environment and the economy; 16% encompassed the environment, the economy and society together, even though it was not done in a dynamic and interlinked way. The aforementioned authors [76] conclude that there is a hierarchy underlying the sustainability dimensions, the environmental one being the most assessed, followed by the economy, and society barely being mentioned.

In April 2019, a paper was published reporting techno-economic analysis along the SLCA. Sadhukhan et al. [77] investigated the avoidance of social impacts when transitioning from an industry based on animal protein, plant sugar, and marine salt into extracting proteins, sugar, and salt from microalgae. Their methodology follows the UNEP/SETAC guidelines, considering as social indicators: "labour rights and decent work; health and safety; human rights; governance; and community infrastructure and underneath twenty-two sub-themes" ([77], p. 2649). These indicators go beyond the impacts on human health, addressing issues of gender and race, social conflicts, infrastructure in the local community and so on [27].

As noted, LCA is focused on the evaluation of processes that are already carried out (an impact assessment), while GC is focused on research and design, in which the processes are generally exploratory and not yet applied on a large scale, which makes it difficult to evaluate by LCA [78]. This may pave the way for important interactions between GC and society if democratic and constructive evaluation processes are proposed, rather than focusing on post facto evaluation. This is the case of constructive technology assessment [79,80], an approach that provides broad social participation in the design process of technologies in which the public contributes to the establishment of a socially committed research agenda and the creation of social indicators relevant to all stakeholders, enabling a more effective technological adherence, representativeness of objectives and plurality of values. This can be an interesting path for GC to tread, expanding its agents and members beyond industrial sectors, government, and academics, including society as a strong partner to promote social transformations and environmental preservation.

The issue is complex and extrapolates the field of expertise of (green) chemists. It is necessary that sustainability metrics researchers and sociologists seek together the construction of acceptable and effective social indicators to evaluate the sustainability efficacy of (green) chemistry. It is also important that these indicators emphasize the need to strengthen GC dialogue with the general population and

decision-makers. GC needs to expand the role of its members, extrapolating the scope of the academy and provoking social mobilization to generate the popular pressure necessary to put into practice the transformations that it intends. Partnerships with environmental movements and other organized civil groups can help to disseminate GC practices and objectives while enabling GC to rethink and enrich its objectives. Chemistry education, hitherto focused on training highly specialized chemical professionals in environmental matters, must also cover the dissemination of GC for the education of non-specialist citizens in the creation of an environmental culture.

Regarding the relationship between GC and social sustainability, a number of problems may be identified:

- GC has no clear relationship with social sustainability;
- The social dimension is subsumed to the economic dimension;
- GC's technological optimism;
- Lack of broad social participation in GC activities;
- Lack of adequate instruments for assessment.

In the following section, we discuss some contributions of chemistry education to address these problems and pursue sustainability.

## 6. The Social Dimension in Chemistry Education

The problem with today's chemistry teaching is that there is too much focus on presenting chemistry substantively (what we know and how we can explain it) and too little effort in teaching chemistry as a creative activity (how we think and what we can do with this form of reasoning) [81]. Within the modernist (or positivist) atmosphere [82], chemistry has developed the image of a set of purely conceptual descriptions (of an ideal form) of reality and has lost its character as a transforming agent [58]; its technological nature [6] was suppressed and its social responsibility was only recently recovered [59].

Educating a responsible citizen is not the same as educating a well-informed individual because it requires more than being able to explain a phenomenon. It is necessary to eliminate the artificial separation between chemistry as content, chemistry as process (research and design), and chemistry as a social agent [81] so that systems thinking is more than thinking about the concepts and processes of chemistry throughout the production chain, but it is also about reflecting on society and its goals, facing chemistry *within* culture and *as* culture [57,82].

Chemical thinking is complex and involves several dimensions that need to be developed together if it is to have an effective and transformative practice. For Sevian et al. [83], chemical thinking involves the knowledge, reasoning, and practices that characterize chemical activity, geared towards the development and application of chemical knowledge for analysis, synthesis, and transformation of matter for practical purposes; for Mocellin [84], the chemical style of reasoning has to do with synthesis, control, and transformation. These are the ontological and technological dimensions of chemistry on which Sjöström [6,82] highlights the epistemological and ethical dimensions. Thus, systems thinking in chemistry may be said to involve four interconnected dimensions (see Figure 1):

- Ontological: chemical theories of description and explanation of reality;
- Technological: procedures of transformation and synthesis-intervention in reality;
- Epistemological: philosophical and sociological perspectives on the production of the chemical knowledge of reality; and
- Ethical: the role of chemistry in society.

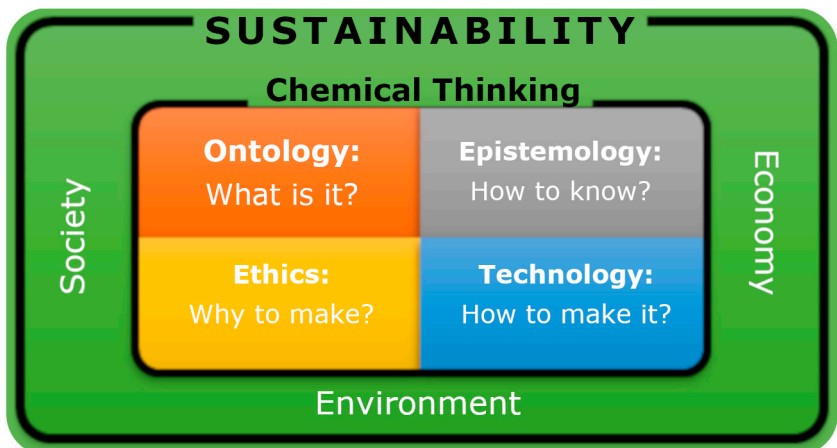

**Figure 1.** Dimensions of chemistry systems thinking inside the context of sustainability.

These dimensions demonstrate the social character of GC, being embedded in the broad context of sustainability, and are the requirements for a reflexive [6,58] education, a *Bildung* education. *Bildung* is a complex concept [85,86] developed in central and northern Europe since the thirteenth century, acquiring an educational status since the 18th century; it describes the movement of incorporation of the individual into culture. For Sjöström [87], *Bildung* is the awareness of the biases that base the opinions and actions of an individual, contrasted with the social context in which one lives, having to do with the competence for self-determination (autonomy, to follow one's own values), constructive participation in society, and solidarity with people limited in their capacity for self-determination and participation.

As discussed above, GC does not openly discuss its relations with society, and yet it is heavily reliant on the pursuit of technological innovations to solve environmental problems, which has been linked to technological optimism. GC teaching is aimed at the training of qualified professionals to develop and implement green technologies, which shows that GC and its teaching primarily develop its ontological (conceptual) and technological (procedural) dimensions, but it is limited in locating the production of green knowledge in the broad social context (epistemological and ethical); it is a systems thinking reduced to the technological context, a "sub-systemic" thinking. Including the social and cultural system in chemical systems thinking, as discussed above, also requires discussing critically what are the present and future needs of humankind and how to include society in planning research programs and building and evaluating technologies. So far, the societal objective of (green) chemistry is not openly discussed [31], but it has been directed to economic growth [82].

There are several ways to relate science, technology, and society, influencing how to approach science teaching. Sjöström et al. [86] propose three approaches in scientific literacy: (1) conceptual, aimed at discovering the "secrets of nature"; (2) contextual, aimed at solving problems of the productive sector through the understanding of science and its application; and (3) critical, "presenting radical solutions to existing (environmental and social) problems and/or new problems beyond the agenda of the (industrial) establishment" ([82], p. 90), considering education as a means to transform individuals and society. In conceptual and contextual strategies, the social purpose of chemistry is not critically reflected; industrial society and technological innovation are considered as the best solution, but it is neither systematically reflected on what the real problem is nor the limits of technologies. This is made by critical scientific literacy, to think science and technology in terms of a critically defined and socially debated social project, contributing not only to the democratization of social processes but also to the enrichment of the knowledge to be created by scientists. The critical approach is that which relates to the concept of critical-reflexive *Bildung*.

Burmeister, Rauch, and Eilks [88] and later authors [6,89] propose four models for GC integration in chemistry curricula to promote sustainability. Model 1 considers the adoption of sustainable practices

in laboratory activities in scientific education, enabling the student to recognize, compare, and reflect on how academia and industries try to minimize the environmental impacts of their activities. Model 2 adds principles of GC and sustainability as topics in theoretical, practical courses, or both, showing the technological advances obtained in the field. In Model 3, controversial questions about sustainability are used for learning *of* science and technology and learning *about* science and technology, making possible the understanding of the arguments involved in the debate and the engagement in the decision-making processes. Model 4 considers the engagement of the whole school and its community in the pursuit of sustainability, extrapolating the scope of teaching activities (teacher-student) and encompassing the role of education within society.

As can be seen, Models 1 and 2 are more related to the conceptual and contextual views of scientific literacy, since they focus on the learning of the ontological and technological dimensions of chemistry. Models 3 and 4 also cover the discussion of the production of chemical knowledge in its broad social context and the need for community engagement to achieve sustainability and, therefore, these models are more adequate for the critical view of scientific literacy and are aligned with the critical-reflexive *Bildung*. Although punctual insertions of GC in chemistry curricula (whether in laboratory practice or on disciplinary topics) are important to disseminate green practices, they are not sufficient to develop effective transformation and education for systems thinking (incorporating ontological, technological, epistemological, and ethical dimensions).

Levinson [90] compared different approaches to science–society relationships in schools, describing the educational purpose of science-technology-engineering–mathematics (STEM) as providing human capital, that of socio-scientific issues (SSI) as development of scientific knowledge needed for socio-scientific reasoning, that of socially acute questions (SAQ) as developing a critical discourse, and that of science and technology education promoting well-being for individuals, societies and environments (STEPWISE) [91,92] as knowledge for action for socio-ecojustice. Besides STEPWISE, other frameworks are also directed to develop socio-ecojustice, like the socio-scientific sustainability reasoning ($S^3R$) model [93], and different socio-critical and problem-oriented approaches of science–technology–society–environment (STSE) studies [94,95] such as the Latin American science, technology, and society (LASTS) [96,97], for instance. In its more socio-critical version (such as STEPWISE, LASTS), it is more than teaching to choose between two or more contradictory alternatives; it is about overcoming the contradiction, unveiling the values and philosophies that underly them, searching for new socio-technical alternatives, and performing actions [55,96].

To promote a critical-reflexive Bildung education, socio-critical SSIs can be used to enable understanding of decision-making processes in society and to engage in critical issues, whether through role-playing exercises, case studies or to unravel the values and interests behind scientific-technological policies and media texts, for example. Another important factor is the use of non-formal education to extrapolate the limits of the school environment and better include society in a scientific-technological discussion.

*Challenges for Social Sustainability and Educational Guidelines*

Table 1 presents some problems in the relationship between GC and social sustainability that may hinder the achievement of full sustainability. In the substantive aspect of social sustainability, GC has not a clear social objective, which is subsumed to the economic dimension. Acknowledging the social genesis of (green) chemistry knowledge in education and discussing an acceptable aim to its practice may be a way to address the epistemological and ethical dimensions of chemical thinking. In the procedural aspect, social sustainability is allegedly achieved by technological innovation, what can be criticized by taking into account technological/entropic inefficiency and the social embeddedness of technology, requiring broad social participation and cultural transformations to achieve sustainability. This calls for a socio-critical *Bildung* education, approaching the entire complexity of chemical thinking in its ontological, technological, epistemological, and ethical dimensions, treating chemistry *in* culture and *as* culture, leading chemistry systems thinking beyond the industrial system and into society.

**Table 1.** Current problems of green chemistry (GC) to address social sustainability (SS) and the respective educational guidelines and chemical thinking (CT) dimensions.

| SS Aspects | Current Problems | Educational Guidelines | CT Dimension |
|---|---|---|---|
| Substantive | No clear relationship between GC and SS Social dimension subsumed to the economic dimension | Contextualizing chemical practices and concepts in their broad social context Discussing the social objectives, the values, and ideologies of chemical practices | Epistemological Ethical |
| Procedural | Technological optimism of GC | Promoting the critical evaluation of technological innovations | Ontological; Technological; Epistemological; Ethical |
| | Lack of broad social participation in GC | Pointing out strategies for broad social participation in scientific, technological and political decision-making processes | Ontological; Technological; Epistemological; Ethical |

## 7. Conclusions

The present analysis has shown that the social dimension is reduced and never explicitly elaborated in GC. The few mentions to the social dimension were related to corporate social responsibility, a self-regulatory initiative of the chemical industries that attempts to change society's perception of chemistry itself. Technological optimism guides GC actions and underlies the belief that the environment and the economy are compatible and they can produce direct and indirect benefits to society (which are not explicitly clarified). At the policy level, GC tries to avoid cost increases due to environmental regulations by creating proactive alternatives. This makes it an elite social movement, targeted at experts and policymakers, ignoring the need for support from other social sectors such as NGOs and environmental movements. Finally, few mentions to social indicators are found, even in more specific tools such as the social life cycle assessment.

Only one analyzed article, published in June 2019, places humanitarian goals above economic progress; integrating benign design, cost reduction, and social dimension into systemic chemical thinking; and underscores the need for qualitative metrics to address this complexity. However, the authors do not make clear the means for GC to achieve its humanitarian goals or to evaluate its performance in this regard so that they make important but still incipient notes.

This qualitative research has the limitation of not covering all GC research, which has tens of thousands of articles. However, it points to a set of important texts (either referenced in textbooks or published in the largest and most important journal in the field), so that the results presented may point to weaknesses and potentialities of this growing field. Further research is needed to explore the relationships between GC, sustainability, and the social dimension, whether in industrial or research practices, in assessment or the concept of systems thinking.

**Author Contributions:** Conceptualization and methodology, all authors; formal analysis, investigation, writing—original draft preparation and visualization, L.M.; writing—review and editing, funding acquisition, and supervision, J.S. and C.A.M.

**Funding:** The corresponding author received funding for a PhD International Exchange at Malmö University in Sweden, Brazil/Capes/PDSE/Process n. 88881.189651/2018-0, and Brazil/CNPq/Process n. 870030/2002-7.

**Conflicts of Interest:** The authors declare no conflict of interest.

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
