# Peer review of "Socio-Problematization of Green Chemistry: Enriching Systems Thinking and Social Sustainability by Education"

_sustainability, doi:10.3390/su11247123_

Round 1

Reviewer 1 Report

This article is centred on a relevant issue: if the Green Chemistry movement has been able to address address the social dimension of sustainability. 

Both the title and the abstract give a clear idea about the article's content, aims and main results. 

The review of the literature is adequate to the discussion of the results and shows knowledge of the research in the area. 

The research questions are clear and the methodology is adequate to the questions and the type of research. However, in this section some aspects need clarification: 1) The selection criteria for the thirteen reviews on Green Chemistry (that mention sustainability issues) are not clear. 2) The process used in the analysis of the publications. How were the social dimensions extracted and grouped? 

The results' section provides relevant information and crosses it with the theoretical framework. The conclusion section is interesting and results from the obtained data.

The numbers' sequence used for different sections needs attention: a) the number 1.1 is used twice; b) the number 1 (and the use of bold) in The Social Dimension in Chemistry Education seems not adequate; c) the number 1 is not adequate for Conclusion section.

Reviewer 2 Report

In this review the authors analyze different aspects concerning the relationship between green chemistry and social sustainability.The paper is well written and the literature is well documented. 

Author Response

Following Reviewer 2's report we have made a language revision throughout the paper. 

Reviewer 3 Report

The manuscript “Green Chemistry and Social Sustainability: Enriching Chemistry System-Thinking by Education” by Marcelino and coworkers focuses on relation between Green chemistry and social sustainability as discussed in published articles.

Although the manuscript report an important ideas and connecting the green chemistry with social and education aspects, several critical points need to be clearly addressed and proved before publication at Sustainability.

Comments:

In line 193, Authors claimed that they discussed 21 papers “retrieved between February and March 2019”. Unfortunately, even this paper is submitted in late 2019, none of the cited papers are published in 2019. It is better to discuss current published literature that published in 2019 rather than relying on retrieving date. First paragraph in the introduction sounds a duplicate of abstract, it needs paraphrasing. Pronoun “we” has been used intensively, it is recommended to paraphrase sentences by minimizing the use of pronouns. Some sentences need to be rewritten with more correct grammar, e.g “We present, then, propositions” line 61 needs rearrangement. Formatting need to be reconsidered. E.g in lines 172-178, double bullets are used. In line 355, this section seems to be odd. It has been numbered 1 and no other numbered sections after. Authors need to consider consistency in titles and subtitles. Conclusion also has number 1 in sequence. All numbering, titles, subtitles need to be reconsidered. Table 1 is recommended to be moved before conclusion and discussed in a separate section

 Finally: major revision should be done though out the manuscript including the sample papers that used for results and discussion before publication

Round 2

Reviewer 3 Report

The revised manuscript sounds ready for publication. 

Formatting is not clear in the word reviewer version, so carefully editing needs to be taken.